# *Apis mellifera* Solinvivirus-1, a Novel Honey Bee Virus That Remained Undetected for over a Decade, Is Widespread in the USA

**DOI:** 10.3390/v15071597

**Published:** 2023-07-21

**Authors:** Eugene V. Ryabov, Anthony J. Nearman, Ashrafun Nessa, Kyle Grubbs, Benjamin Sallmann, Rachel Fahey, Mikayla E. Wilson, Karen D. Rennich, Nathalie Steinhauer, Anne Marie Fauvel, Yanping Chen, Jay D. Evans, Dennis vanEngelsdorp

**Affiliations:** 1Department of Entomology, University of Maryland, College Park, MD 20742, USA; anearman@umd.edu (A.J.N.); anessa@umd.edu (A.N.); faheybrl@umd.edu (R.F.); electra@umd.edu (M.E.W.); krennich@umd.edu (K.D.R.); nsteinha@umd.edu (N.S.); fauvelam@umd.edu (A.M.F.); dvane@umd.edu (D.v.); 2Bee Research Laboratory, USDA-Agricultural Research Service, Beltsville, MD 20705, USAjudy.chen@ars.usda.gov (Y.C.); jay.evans@ars.usda.gov (J.D.E.); 3Bee Informed Partnership, College Park, MD 20742, USA; bensallmann@gmail.com; 4Department of Horticulture, Oregon State University, Corvallis, OR 97331, USA; 5Department of Entomology, University of Minnesota, St. Paul, MN 55108, USA

**Keywords:** *Apis mellifera*, honey bee, honey bee colony losses, RNA virus, insect, pollination service, family *Solinviviridae*

## Abstract

A metagenomic analysis of the virome of honey bees (*Apis mellifera*) from an apiary with high rates of unexplained colony losses identified a novel RNA virus. The virus, which was named *Apis mellifera* solinvivirus 1 (AmSV1), contains a 10.6 kb positive-strand genomic RNA with a single ORF coding for a polyprotein with the protease, helicase, and RNA-dependent RNA polymerase domains, as well as a single jelly-roll structural protein domain, showing highest similarity with viruses in the family *Solinviviridae*. The injection of honey bee pupae with AmSV1 preparation showed an increase in virus titer and the accumulation of the negative-strand of AmSV1 RNA 3 days after injection, indicating the replication of AmSV1. In the infected worker bees, AmSV1 was present in heads, thoraxes, and abdomens, indicating that this virus causes systemic infection. An analysis of the geographic and historic distribution of AmSV1, using over 900 apiary samples collected across the United States, showed AmSV1 presence since at least 2010. In the year 2021, AmSV1 was detected in 10.45% of apiaries (95%CI: 8.41–12.79%), mostly sampled in June and July in Northwestern and Northeastern United States. The diagnostic methods and information on the AmSV1 distribution will be used to investigate the connection of AmSV1 to honey bee colony losses.

## 1. Introduction

Honey bees, *Apis mellifera*, are the principal managed pollinators of a large number of plants in both farmed and natural systems worldwide [1]. Honey bees are subject to diseases caused by parasites and pathogens, including viruses [2], which cause increased colony losses, posing a risk to pollination service that is essential for the production of many crops. For over a decade, high rates of honey bee colony losses have been reported in many regions of the world, notably in the United States [3]. Current research, which aims to unravel the impact of viruses on honey bee health and colony survival, largely relies on the detection and quantification of known viruses [4]. Despite an extensive research effort, there is no complete understanding of the connection between pathogen loads and colony health. Among the reasons for this, one could be incomplete knowledge of virus diversity, there is a possibility that unknown viruses or new variants of known viruses, which are not targeted by existing screening programs, may contribute to colony losses. Therefore, identification of all components of the honey bee virome is important for the prediction of viral impacts on honeybee health and the development of virus mitigation measures.

Metagenomic high-throughput sequencing became a universal method of discovery of novel viruses and virus variants in invertebrates using field-collected samples [5]. In this study, we used this approach to screen for viruses in honey bees from a beekeeping operation that experienced high rates of unexplained colony losses. We identified a novel RNA virus, a member of the family *Solinviviridae* [6], which we named *Apis mellifera* solinvivirus-1 (AmSV1). Little is known about the effect of solinviviruses on their hosts, but a negative impact on an invertebrate host was shown for the viruses of this group infecting fire ants, SINV3 [7,8], as well as the most important cultured shrimp species, in which infection of *Penaeus vannamei* solinvivirus (PvSV) was associated with unusually high mortality [9]. Those examples highlight the potential of solinvivirus species to negatively impact at least some of their invertebrate hosts in other systems.

We further investigated the replication of AmSV1 in honey bees and analyzed its geographic, seasonal, and historic distribution in the United States and worldwide. Results of our study suggest that AmSV1 should be further investigated to determine its impact on honey bee health and colony survival.

## 2. Materials and Methods

**High-throughput sequencing de novo virus assembly, sequence verification, and bioinformatic analysis.** For metagenomic sequencing, live worker honey bees were collected from an apiary with a historically high level of colony losses, located in Oregon, in June 2018. This sampling was conducted by the Bee Informed Partnership Tech Transfer Team monitoring program as part of a larger effort aimed at determining the putative cause of unexplained high rates of loss experienced by some commercial beekeepers in the Pacific Northwest and Northern Plain states of the United States [10].

Total RNA was extracted from frozen worker honey bee pools by using TRIzol reagent (Thermo Fisher, Cleveland, OH, USA), and it was further purified using RNeasy columns according to manufacturers’ instructions. Poly(A) fractions were isolated and, then, sequenced by RNA-seq technologies on an Illumina HiSeq2000 platform at the Genomics Resource Center (GRC), University of Maryland, Baltimore, MD to produce libraries of approximately 50 million paired-end 100 nt reads. The reads were checked for quality and trimmed by the Genomics Resource Center prior to being mapped to the *Apis mellifera* honey bee transcriptome (OGSv3.2) [11] with the sequences of all known viruses infecting honey bees and *Varroa* mites with bowtie2 [12]. We detected reads belonging to deformed wing virus-A, Sacbrood virus, black queen cell virus, Israeli acute paralysis virus Apis rhabdovirus-1, Apis rhabdovirus-2, and Lake Sinai virus. The un-aligned reads were then used for de novo assembly using Spades software [13]. The contigs 6–10.6 kb long and their Open Reading Frames (ORF) were analyzed and compared against sequences deposited to GenBank by the BLAST search tool [14]. The search for similarities was done using tblastn, within the NCBI nr/nt database, and the cutoff E-value of 0.0 was used. This search showed that the long ORF of some contigs coded for novel proteins, showing 17–34% similarity to the proteins of the recognized and putative members of the family *Solviniviridae*, order *Picornavirales*, which allowed for the identification of sequences of the novel virus, which was tentatively named AmSV1. 

A set of oligonucleotide primers, designed according to the AmSV1 contig (Appendix A), was used to produce a series of overlapping RT-PCR fragments covering the viral genome with the RNA extract in which AmSV1 was discovered. The RT-PCR fragments were sequenced using the Sanger method, and the confirmed sequence was submitted to GenBank under accession number OQ540582 (Appendix A). 

The alignment of inferred protein sequences was performed using the Clustal Omega tool [15]; then, phylogenetic analysis and bootstrapping were performed using the PHYLIP package [16]. 

To investigate the worldwide distribution of AmSV1, publicly available honey bee RNA-seq libraries were screened for the presence of AmSV1. The AmSV1 sequence produced in our study, GenBank accession number OQ540582, was used as a reference sequence with Bowtie2 alignment tool [12] and samtools [17], and the samtools mpileup output was essentially used to calculate Shannon’s diversity, as in Ryabov et al., 2017 [18]. 

**US-wide honey bee sampling.** Live worker honey bees were collected from apiaries involved in the National Honey Bee Disease Survey (NHBDS) [4]. In total, 930 apiary-level samples from the survey years—2010, 2014, and 2021—were included in this study. In each apiary, 7 or 8 colonies were usually sampled per apiary (in the year 2021, mean ± standard deviation, 7.71 ± 1.07) to obtain a combined apiary-level sample. The bees were immediately frozen in dry ice, shipped frozen to the University of Maryland, College Park, and stored at −80 °C prior to RNA extraction. For each sampled apiary, in-field standard colony health metrics were noted, including the number of sampled colonies with overt expression of common brood disease and their queen status. The apiary-level analysis of *Varroa*, *Nosema* sp., and virus prevalence and loads were quantified using methods standard in processing NHBDS samples, as described in Traynor et al. (2016) [4]. These data were entered, in a standardized format, into a secure, custom-designed online relational database. A publicly accessible interface is available to this database, which provides aggregated summaries and de-identified data downloads at https://research.beeinformed.org/state_reports/ (accessed on 1 March 2023).

**Virus purification and pupal injection experiment.** To obtain partially purified preparations of AmSV1, we used adult worker honey bees from the apiary samples, which had tested positive for AmSV1, were collected in an apiary in New York State in the summer 2021, and were stored at −80 °C. Entire individual bees (n = 10) were homogenized with 25 mL of ice-cold phosphate-buffered saline (PBS) solutions, with pH of 7.4, and they were subjected to low-speed centrifugation at 10,000× *g* for 10 min at +4 °C to remove debris. The supernatant was layered over 10 mL of PBS-buffered 20% sucrose, and the tubes were spun at 28,000 rpm, for 4 h at 18 °C, in a bucket SW-28 rotor (Beckman, Brea, CA, USA). The pellet was resuspended with 500 μL of PBS, incubated on ice for 30 min with periodic vortexing, and then centrifuged for 5 min at 10,000 rpm to remove aggregates. The supernatant was filtered through a 0.21 micrometer nylon syringe filter (Thermo Fisher, Cleveland, OH, USA) and stored at −80 °C prior to infection. An aliquot of the filtered extract was used to extract RNA using RNeasy kit, and it was determined with RT-qPCR that the extract contained 3 log_10_10 AmSV1 genome equivalents (GE) per microliter. Honey bee pupae at the early pink eye stage that were obtained from a Florida colony, which was free of AmSV1, were injected intra-abdominally with 1 μL of the AmSV1 preparation, 10^3^ GE, and resuspended in 5 μL PBS. The injected pupae were sampled immediately after injection (defined as Time 0) and after 3 day incubation at 32 °C (3 dpi). Total RNA was extracted from the pupa individually, and then, it was used for the RT-qPCR detection of AmSV1 and for the specific detection of the negative strand of AmSV1. 

**Assessment of prevalence and load of AmSV1.** Loads of AmSV1 were quantified by two-step reverse-transcription quantitative PCR (RT-qPCR). For the assessment of the spatio-temporal distribution of AmSV1 in the US beekeeping operations, we used archived apiary-level honey bee RNA samples (n = 930), isolated from pools of 50 worker bees collected as part of the USDA-APHIS National Honey Bee Disease Survey [4], in which levels of other most common honey bee viruses, such as *Nosema cerana*, as well as levels of the *Varroa* mite infestation, were determined. The virus levels were also quantified in honey bee pupae injected with viral inoculum and in the body sections of individual frozen worker bees from the identified AmSV1-positive apiaries. For this, pupae or sections of adult worker bees were individually placed into tubes with approximately 0.1 mL of ceramic (zirconium silicate) beads that were 1 mm in diameter, with 1 mL of TRIzol reagent (Thermo Fisher), and they were immediately homogenized at 30 Hz in a Precellys 24 High-Powered Bead Mill Homogenizer. Further RNA isolation steps were carried out with TRIzol reagent, according to the manufacturer’s instructions. The total RNA pellets were washed twice with 75% ethanol, air-dried, and dissolved with 100 mL of RNAse-free water. Nanodrop was used to measure the concentration of RNA to calculate volumes containing 2.5 mg of total RNA, which was used to produce cDNA with iScript Advanced cDNA Synthesis kit (Bio-Rad, Hercules, CA, United States), according to the manufacturer’s instructions. The cDNA samples were used to determine AmSV1 copy numbers by quantitative PCR with the oligonucleotide primers, qPCR-AmSV1-For (5′-GCATTTACAACGAACCAGTAAC-3′) and qPCR-AmSV1-Rev (5′-CATTGAATAGTGTTGAGTGCTGTG-3′), targeting the region 5541 to 5658 in the AmSV1 genome using SsoAdvanced Universal SYBR Green Supermix (Bio-Rad, Hercules, CA, United States). A series of 10-fold dilutions of the RT-PCR fragment, containing the target for the AmSV1 qPCR primers, was used to establish a standard curve by plotting Ct values against the log-transformed concentrations. The Ct values showed linearity (R2 = 0.9987) and provided an amplification efficiency of 1.92. The detection threshold for AmSV1 was 2.8 log10 genome equivalents per bee. For the detection of the negative-strand RNA of AmSV1, total RNA extracted from honey bees was used to produce cDNA with the tagged forward primer, corresponding to the region 5541–5562, in positive polarity (5′-CTTGGTTAGCTGTGTTGCAGTTGCATTTACAACGAACCAGTAAC-3′, the tag sequence is underscored). Then, the resulted cDNA was used as a template in PCR reaction with the primer targeting the tag (5′-CTTGGTTAGCTGTGTTGCAGTTG-3′) and the reverse primer AmSV1-Rev. 

**Statistical analysis.** Statistical analyses were conducted using R Software version 4.2.1 (R Foundation for Statistical Computing, Vienna, Austria) [19], and figures were produced using the packages ggplot2 [20]. Associations between AmSV1 and sample month or climate zone were determined using logistic regression, with AmSV1 positive months or climate zones as the independent variable and the apiary level AmSV1 presence or absence as the dependent variable. Models were generated using the base R *glm* function, and multiple comparisons were calculated with the multcomp package [21]. To determine associations between AmSV1 and either parasite prevalence or field level observations at the time of sampling, odds ratios were calculated using the epitools package [22].

## 3. Results

### 3.1. Apis mellifera Solinvivirus 1 Identification and Sequence Analyses 

We carried out metagenomic the virome analysis of adult worker bees collected from an apiary that had a high rate of unexplained colony loss [10]. An RNA-seq paired-end library of 57144487 paired-end 100 nt reads was produced for the poly(A) enriched fraction of total RNA extracted from 200 bees from 4 colonies of this apiary (NCBI BioProject and SRA data: PRJNA945618). After the removal of reads corresponding to the honey bee transcriptome, *Varroa*, and bee viruses known to date, de novo assembly analysis was carried out. We assembled a 10.6 kb RNA virus sequence containing a single 10.5 kb-long open reading frame (ORF), which encoded a protein showing the highest similarity (17% to 34%) to the proteins of the recognized and putative members of the family *Solviniviridae*, order *Picornavirales* (Figure 1e) [6]. The family *Solviniviridae* includes the genus *Invictavirus* (classified species *Solenopsis invicta* virus-3, SINV3), which has two open reading frames expressed by a translational -1 frameshifting, and the genus *Nyfulvavirus* (classified species *Nylanderia fulva* virus 1, NfV1), which has single extended ORF coding for the same domains [7,23]. Based on sequence similarity and genome organization, we concluded that the novel sequence is the genomic RNA of a previously unknown virus in the genus *Nyfulvavirus*, and therefore, we named it *Apis mellifera* Solvinivirus-1 (AmSV1). The novel genome verified by Sanger-sequencing of the overlapping RT-PCR fragments was deposited to GenBank under accession number OQ540582 (Appendix A). 

Alignment of the paired-end NGS reads from the library showed coverage ranging between 100 and 1000 reads for most of the genome, from the 5′ end for approximately 8.5 kb and up to 5000–50,000 reads in the 3′ section, corresponding to the putative subgenomic RNA for the expression of the structural genes [6] (Figure 1b). 

The full genome of AmSV1 is 10,632 nt (excluding the 3′ terminal poly A sequence). The AmSV1 ORF codes for a 3506 amino acid (aa) polyprotein, in which we identified conserved sequence motifs typical for the non-structural domains of picorna-like viruses, including the RNA helicase (aa positions 630–750), protease (aa positions 1600–1660), RNA-dependent RNA polymerase (aa positions 2100–2450), followed by VP1, and the Calicivirus-type jelly-roll fold capsid protein domain (aa positions 2705–2864), Figure 1a, Appendix A. It is believed that 180 copies of the jelly-roll structural viral protein (VP) 1 subunits form the shell of icosahedral virus particle, and the VP2 forms a protruding domain. 

The sequenced apiary-level AmSV1 population showed high population diversity (Shannon’s diversity index, Figure 1c). We found that the tested NGS library had 419 polymorphic nucleotide positions, with alternate allele proportions exceeding 3%. This nucleotide variation results in 63 amino acid changes (Figure 1d). While polymorphic nucleotide positions were distributed throughout the entire genome, the alternate amino acid positions were mostly present in the regions of the polyprotein linking the conserved domains, which no variation in the RdRp domain. Notably, very few amino-acid changes were observed in the structural proteins and none of them in the JR (VP1) domain (Figure 1d).

We carried out a phylogenetic analysis of the complete polyprotein sequences of AmSV1, including both classified solinvivirus species and unclassified invertebrate viruses, as well as potential solinviviruses, which had the same genome organization. It was found that AmSV1 was grouped in the same clade as both classified solinviviruses, SINV3 and NfV1, with 96% bootstrap support. AmSV1 showed highest similarity with the sequence MN918666 from an unknown insect species and *Diabrotica virgifera* virus 2, with 36.8% and 35.5% shared identity, respectively. Both classified Solinviviruses infect ants, but potential members of this family also infect beetles, mosquitoes, aphids, Asian bees, *Apis cerana*, and shrimps. AmSV1 showed 29.2% identity with one of several closely related solinviviruses found in *Apis cerana*, *MZ822083,* which was not clustered in the same clade as AmSV1 (Figure 1e). 

### 3.2. Distribution of AmSV1 in Infected Worker Honey Bees Indicates Systemic Infection

Quantification of AmSV1, which loads the body segments of individual worker bees, was performed as a first step to determine that the virus replicates in honey bee tissues and is not just an accumulation of viral material in the digestive tract after oral acquisition from an unknown source. We used adult worker honey bees (n = 16) that were collected from two AmSV1-positive apiaries, from West Virginia and New York states in 2021 (see below) and stored frozen at −80 °C, to investigate the distribution of AmSV1 in infected bee bodies. For this purpose, we cut individual frozen worker bees into three sections—head, thorax, and abdomen (Figure 2a)—and quantified the virus loads in each of these segments using RT-qPCR (Figure 2b). 

We found that similar loads of AmSV1 were present in all three bee body parts (ANOVA, df = 2, F = 1.73, *p* = 0.1889). The strongest correlation was found between the viral loads in heads and thoraxes (R = 0.72, *p* = 0.0017), as well as between thoraxes and abdomens (R = 0.52, *p* = 0.032) (Figure 2b). Notably, loads of AmSV1 in heads and thoraxes exceeded those in abdomens in some tested bees, which showed that AmSV1 was not limited to bees’ digestive tracts, which are mainly located in abdomen [24], as it would be in the case for predominantly abdominal detection. Thus, distribution of AmSV1 in bee bodies indicates that the virus causes systemic infection in adult workers, i.e., it spreads through different organs and tissues. This implies that it replicates in honey bee cells. The presence of AmSV1 in 15 of 16 randomly chosen worker bees in infected colonies demonstrated that the virus is widespread among workers in the infected apiaries.

### 3.3. Replication of AmSV1 in Honey Bee Tissues in Pupal Injection Experiments

Frozen worker bees from an AmSV1-positive colony, collected in 2021 (see below), were used for the virus purification by ultracentrifugation through a sucrose cushion. The virus particles, approximately 10^3^ genome equivalents (GE), were injected intra-abdominally into the haemocoel of honey bee pupae. A randomly chosen half of the injected pupae was harvested and frozen immediately (“AmSV1 Time 0 dpi”), while the remaining pupae were incubated at +33 °C and harvested and frozen three days later, which allowed for the development of viral infection (“AmSV1 3 dpi”0). The quantification of the AmSV1 loads by RTqPCR, mostly genomic positive strand RNA, showed a significant increase in the AmVS1 levels in the “AmSV1 3 dpi” pupae compared to those sampled immediately after virus injection (Figure 3a). No AmSV1 was detected in the buffer-injected pupae 3 days post-injection (“PBS 3 dpi”). A search for the negative strand of AmSV1 RNA, a replication intermediate, was carried out in the virus preparation and the samples used for AmSV1 quantification. The specific PCR product was only detected in the virus-injected pupae 3 days after injection but not at the Time 0 (Figure 3b), confirming that replication of AmSV1 took place in the injected pupae. 

### 3.4. Prevalance and Loads of AmSV1 in the USA Apiaries

We used RTqPCR to determine the prevalence and loads of AmSV1 in apiary-level samples of worker bees collected across the United States by NHBDS and stored frozen at −80 °C. An analysis of RNA extracted from samples collected in 2021 (n = 794) showed an overall AmSV1 prevalence of 10.45% (95% Confidence Interval (95%CI): 8.41–12.79%), (Figure 4b). 

We also found that the prevalence and load of AmSV1 peaked in the months of June and July in 2021, reaching 20.7% (95% CI: 15.0–27.8%, n = 109) and 16.8% (95% CI: 11.2–24.5%, n = 99), respectively. The prevalence of AmSV1 dropped to 5% (95% CI: 2.6–9.6%, n = 106) in September 2021. There were very few samples collected during the winter months. All 3 samples with detectable levels of AmSV1 collected during the winter months came from Texas and Florida (marked in Figure 4b,c). Minor differences between climate zones were also detected, with the highest being found in the Northeast zone at 23.9% (95% CI: 16.4–32.8%) and the Northwest zone at 18.8% (95% CI: 10.4–30.1%) in the United States (Appendix A). 

A retrospective analysis of select archived RNA NHBDS samples, collected in 2010 (n = 73) and 2014 (n = 63), found AMSV1 prevalent in 13.79% (95% CI: 6.77–23.75%) and 14.06% (95% CI: 6.64–25.02%) of tested samples, respectively (Figure 2a). 

We used odds ratios (OR) to describe relationships between AmSV1 and other honey bee parasites quantified by the NHBDS, including eight viruses, the microsporidium *Vairimorpha ceranae* and the *Varroa destructor* mite (Figure 5a, Appendix A). We found that AmSV1 was 73% less likely to be present in apiaries with detectable levels of DWV-A (OR = 0.578, 95% CI: 0.353–0.960, *p* = 0.042, Figure 5a). The lack of significant connection between AmSV1 loads and *Varroa* infestation levels (OR = 0.740, 95% CI: 0.356–1.405, *p* = 0.806) suggests that this virus may not be transmitted by *Varroa destructor*. 

Odds ratios were also calculated for AmSV1 and apiary field health observations, including the presence of overt brood diseases and queen status at the time of sampling (Figure 5b, Appendix A). Apiaries with detectable levels of AmSV1 were nearly twice as likely to contain queenless colonies (OR = 1.945, 95% CI: 1.083–3.381, *p* = 0.032, Figure 5b). No other observations drew significant associations from prevalence data. 

Similar to prevalence observations, we calculated Pearson’s correlations between the AmSV1 load and either the load of other viruses or apiary field observations. A positive relationship between AmSV1 viral loads and field observations, for the number of colonies in an apiary that has worker bees with deformed wings, was detected (*r* = 0.813, 95%CI: 0.253, 0.965, *p* = 0.014). No additional significant correlations were found amongst load data. 

### 3.5. Worldwide Distribution of AmSV1 

We examined publicly available NGS libraries from NCBI for the presence of AmSV1 reads. In total, we screened 542 *Apis mellifera* RNA-seq libraries from 6 countries (Brazil n = 61, China n = 59, Germany n = 68, Turkey n = 96, United Kingdom n = 159, United States n = 99) across all years from 2013 to 2021, except 2020, where no libraries were available (Appendix A). 

## 4. Discussion

In this study, by using high-throughput RNA sequencing and de novo assembly, we investigated the virome of a U.S. honey bee apiary showing historically high colony losses [10]. We discovered a novel positive-strand RNA virus, a putative member of the family *Solinviviridae,* which we named *Apis mellifera* solinvivirus 1 (AmSV1). 

A retrospective analysis of archived honey bee samples revealed that AmSV1 is prevalent in just over 10% of samples collected in 2021, and it has been present in the United States since at least 2010. We provided evidence that AmSV1 is capable of replicating in artificially injected honey bee pupae, as manifested by an increase in AmSV1 RNA following an injection of the purified virus preparation (Figure 4a) and by the accumulation of negative strand ofAmSV1, which further confirmed the replication of this virus in honey bee cells (Figure 4b). 

The analysis of high-throughput sequencing data in the Oregon apiary library, where AmSV1 was discovered, showed that the population of this virus had a high level of diversity (Figure 1c,d). The level of genetic diversity was similar to that observed in 2015 for the U.S. populations of deformed wing virus (DWV) type A, a virus that was present in the U.S. honey bees since at least 2004 [18]. AmSV1 was, however, more diverse than DWV type B [25], which started spreading in the United States in 2010. This result, together with our detection of this virus in honey bee samples collected in 2010, suggests that AmSV1 has been circulating in the United States for a considerable time. 

We demonstrated that AmSV1 replicates in honey bees by using an approach previously applied to investigate replication of DWV in honey bees [26], namely the quantification of viral RNA in different honey bee body parts and detection of a replicative intermediate, the viral negative RNA strand in virus injected bees. We found that AmSV1 was present in all bee body segments (heads, thoraxes and abdomens) of individual worker honey bees collected from the virus-positive apiaries (Figure 3). The detection of similar levels of AmSV1 in heads, thoraxes and abdomens of infected individuals, indicates that this virus is infecting worker bees systemically reflecting replication in honey bee cells. The loads of AmSV1 in individual worker bees, 10^6^ GE per insect (Figure 2a), were similar to the maximum per bee loads observed for the apiary-level samples (Figure 3b,c). We further confirmed replication of AmSV1 in honey bees in a pupal injection experiment, in which a significant increase in AmSV1 RNA was observed after three days post injection, and was accompanied by the accumulation of the negative-strand AmSV1 RNA (Figure 3b). 

We investigated the spatio-temporal distribution of AmSV1 in honey bees across the United States by using honey bee samples collected for the National Honey Bee Disease survey [4]. AmSV1 prevalence and load were quantified by RTqPCR bee RNA preparations from apiary-level worker bee samples (n = 930) collected across the United States in 2010, 2014, and 2021. An analysis of all apiaries sampled in 2021 (n = 794) showed that the virus is currently widespread in the United States, and it had highest incidence in the Northeast and Northwest regions of the U.S. mainland (Figure 4a, Appendix A). 

The analysis of historic samples, which analyzed a subset of available samples, showed that AmSV1 has been circulating in the United States since 2010 (Figure 4b), further supporting our supposition that AmSV1 is endemic in U.S. honey bee populations. This might explain the high genetic diversity of AmSV1 observed in the sequenced apiary (Figure 1c,d). In 2021, the year for which we completed the cross-sectional survey, we found that AmSV1 prevalence varied during the year, reaching the highest levels in June and July—20.7% (95% CI: 15.0–27.8%) and 16.8% (95% CI: 11.2–24.5%), respectively—before significantly decreasing to 9.4% (95% CI: 5.33–0.16.1%) in August and 5% (95% CI: 2.6–9.6%) in September (Figure 4b,c). 

We used publicly available honey bee RNA-seq libraries produced for the bees, sourced worldwide, to investigate the global distribution of AmSV1. The virus was not detected outside the United States, suggesting that it might spread to honey bees from unknown insect hosts in the United States. It is also possible that AmSV1 was not detected because the majority honey bee high-throughput transcriptome studies involved honey bees originating from managed colonies without health issues. 

The apiary-level samples used for the quantification of AmSV1 were investigated for the prevalence of the most common honey bee viruses in the United States [4], as well as *Vairimorpha ceranae* and *Varroa destructor* mites (which vector a number of honey bee viruses) [27] and other measures of honey bee health, including overt diseases and queen condition, which were recorded at the time of sampling [28]. This permitted us to quantify correlative relationships between other honey bee parasites and in-hive conditions (Figure 5a,b). We found that the levels of only one virus, DWV-A, showed significant connection with AmSV1. We observed a negative correlation between DWV-A and AmSV1 (odds ratio 0.578, *p* = 0.042), suggesting that AmSV1 and DWV-A are competing with each other. While this hypothesis would require further testing, it might explain the reduction in AmSV1 levels in September and October, the months when DWV-A reaches maximum levels. 

Routes of AmSV1 transmission, including the possible role of *Varroa* mites, also require investigation. So far, we found no significant connection between AmSV1 prevalence, either *Varroa* mite infestation, or signs of parasitic mite syndrome (Figure 5a,b), suggesting that AmSV1 is not transmitted by *Varroa* mites. 

The significant positive connection between AmSV1 and the presence of queenless colonies in sampled apiaries (Figure 5b, odds ratio = 1.945, *p*-value = 0.032) suggests a link between the virus and poor queen health. Beekeepers have long known that “failing queens” are often replaced with a new queen [29]. Not all attempts to replace a colony’s queen are successful; as such, queen failure is identified by U.S. beekeepers as a leading cause of colony loss [28,30]. We discovered AmSV1 in an apiary with historically high levels of colony losses, which might suggest increased queen failure rates. In this respect, the effect of AmSV1 on honey bees might have a similar impact to the effect another solinvivirus, SINV3, has on its fire ant host. It was shown that oral infection of fire ant colonies with SINV3 resulted in an approximately 10-fold reduction in the number of eggs in the ant queen ovaries, which likely led to a significant reduction in the number of eggs laid by queens and a decrease in brood and worker numbers [31]. While experimental colony inoculation would be required to test whether AmSV1 leads to queen failure, preliminary data suggests that this could be the case. This warrants further investigation of the possible pathological impact of AmSV1 on honey bees and, in particular, on the fecundity of honey bee queens. Solinviviruses remain poorly characterized, and future work on AmSV1 biology could reveal possible ecological impacts of other members of this virus group. 

## Figures and Tables

**Figure 1 viruses-15-01597-f001:**
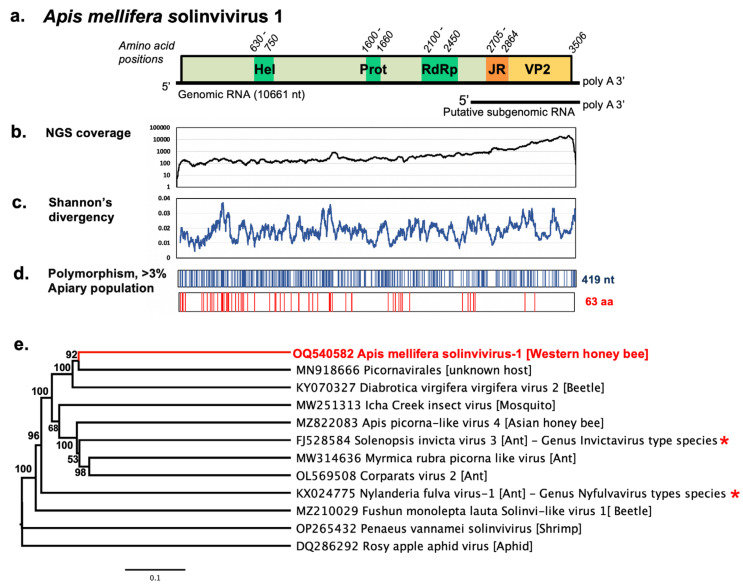
The organization of *Apis mellifera* solinvivirus-1 genomic RNA. (**a**) The schematic representation of AmSV1 genomic RNA (GenBank accession number OQ540582). The position of the main ORF and putative protein domains is shown. Amino acid positions of the protein domains are indicated above the ORF. Non-structural proteins: Hel—helicase, Prot—3C protease, structural proteins: JR—jelly roll domain of structural viral protein (VP)1, VP2; (**b**) NGS coverage of the AmSV1 genome. Genetic diversity of AmSV1 apiary-level population used for NGS analysis: (**c**) Shannon’s diversity profile (sliding window average for 100 nt positions); (**d**) distribution of polymorphic nucleotides (n = 419) and amino acids (n = 63), showing an alternate allele exceeding 3% in frequency in the apiary-level NGS library. (**e**) The maximum likelihood phylogenetic tree was generated based on the full-length protein sequences of AmSV1, as well as classified (marked with asterisk) and putative solinviviruses. For SINV3 and RAAV, the sequences of -1 translational frameshift proteins (ORF1-ORF2 fusions) were used. Bootstrap values above 50%, generated from 1000 replications, are shown to the left of corresponding nodes. The bar indicates a 10% sequence difference.

**Figure 2 viruses-15-01597-f002:**
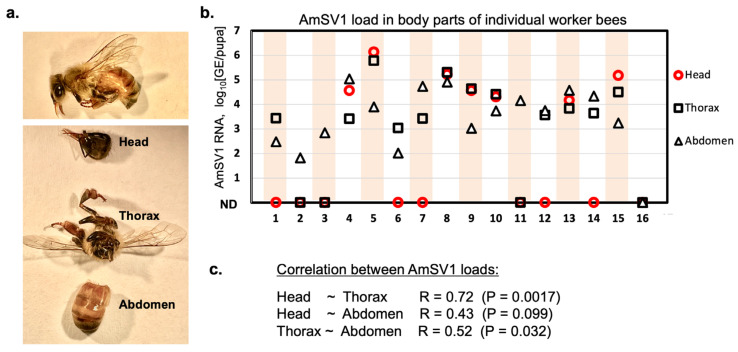
Accumulation of AmSV1 in different body parts of individual adult honeybee workers from AmSV1 positive apiaries. (**a**) Sections of a frozen worker honey bee used for RNA extraction. (**b**) AmSV1 loads in the head, thorax, and abdomen of 16 worker bees. (**c**) Correlation between AmSV1 loads in head, thorax, and abdomen.

**Figure 3 viruses-15-01597-f003:**
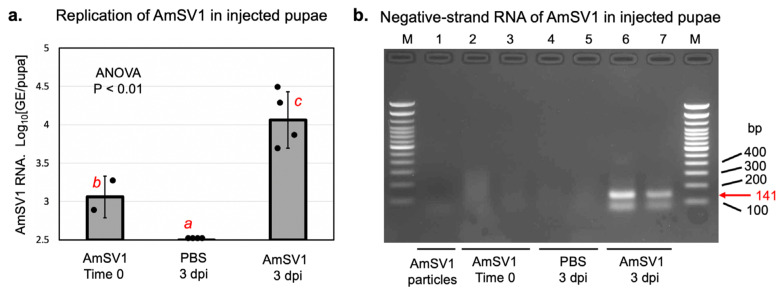
Pupal injection experiment. (**a**) The quantification of AmSV1 genome equivalents (GE) in the pupae, injected with partially purified AmSV1 preparation (AmSV1) or buffer control (PBS), which were sampled immediately after injection (Time 0) and after 3 days of incubation at +33 °C (3 dpi). Quantification was carried out by qPCR using cDNA-generated random primers, allowing the detection of AmSV1 RNA of both polarities. Dots indicate levels of AmSV1 in individual pupae. Significantly different levels of AmSV1 RNA are indicated by different red letters (ANOVA *p* < 0.01). (**b**) Specific detection of negative-strand RNA, a virus replicative intermediate, in virus preparation (lane 0) used for injection. The pupae was injected with partially purified AmSV1 preparation (AmSV1) and sampled immediately after injection, Time 0 (lanes 2 and 3, individual pupae), or after 3 day incubation at +33 °C, with 3 dpi (lanes 6 and 7, two pools of 2 pupae). Control, buffer-injected pupae—PBS—were sampled 3 days after injection (lanes 4 and 5, two pools of two pupae). M, DNA ladder, base pairs (bp). The cDNA was produced using tagged forward primer, PCR amplification was carried out with the primer that was identical to the tag and reverse primer. The arrow marks the position of the expected 141 bp RT-PCR product.

**Figure 4 viruses-15-01597-f004:**
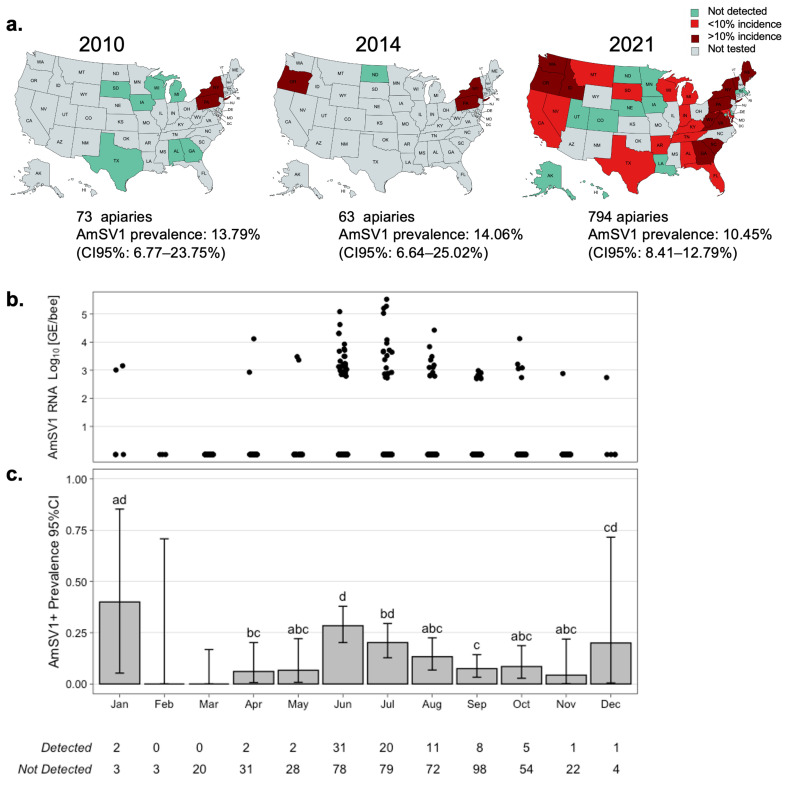
Spatio-temporal distribution of AmSV1 in the USA. (**a**) The distribution of AmSV1 in the US apiaries in 2010, 2014, and 2021. (**b**) Monthly loads (0 = not detected, below 2.8 log_10_ GE/bee) and (**c**) monthly distribution of AmSV1 prevalence and loads for the year 2021.

**Figure 5 viruses-15-01597-f005:**
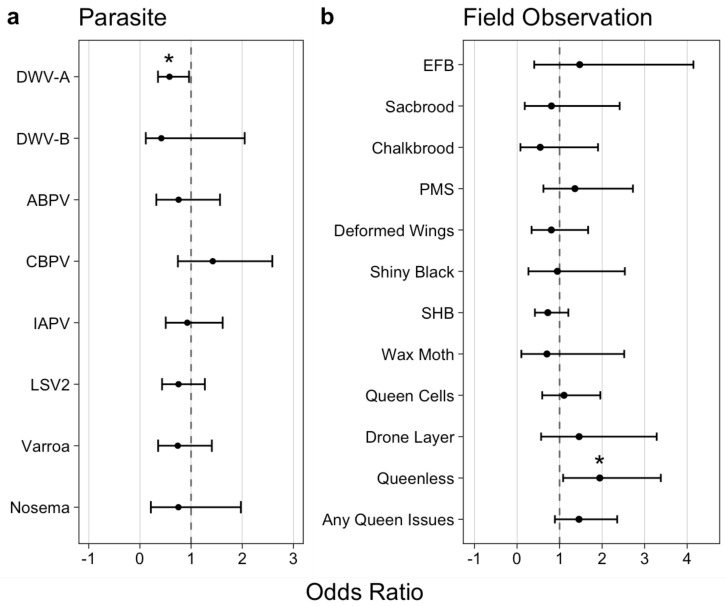
Connections between the prevalence of AmSV1 and (**a**) other honey bee parasites, including (**b**) field apiary observations in the U.S. 2021 apiaries. In total, 794 apiary-level samples were tested. Dashed line at OR = 1 represents the null hypothesis that AmSV1 does not associate with listed measures. DWV–A—deformed wing virus, type A; DWV-B deformed wing virus type B; ABPV—acute bee paralysis virus; CBPV—chronic bee paralysis virus; IAPV—Israeli acute paralysis virus; KBV—Kashmir bee virus; LSV2—Lake Sinai virus; Varroa—ectoparasitic mite *Varroa destructor*; Nosema—*Vairimorpha ceranae*. EFB—European foulbrood (caused by bacterium *Melissococcus plutonius*), Sacbrood–caused by sacbrood virus, Chalkbrood–fungal disease of honey bee brood caused by fungus *Ascosphaera apis*, PMS—Parasitic Mite Syndrome (caused by the mite *Varroa destructor*), Deformed Wings–could be caused by DWV, Shiny Black—hairless bees, SHB—infestation with small hive beetle (*Aethina tumida*), Wax Moth—infestation with wax moth (*Galleria mellonella*), Queen Cells presence, Drone Layer–queen lays unfertilized drone eggs, Queenless—queen is absent in at least one of sampled colonies, Any Queen Issues—combined Queen Cells, Drone Layer, and Queenless. A significant *p*-value is marked with an asterisk.

## Data Availability

The nucleotide sequence of AmSV1 is deposited to GenBank, accession number OQ540582. High-throughput sequence data is deposited to NCBI GenBank, BioProject PRJNA945618, BioSample accession number SAMN33791926.

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
