# Peer review of "Apis mellifera Solinvivirus-1, a Novel Honey Bee Virus That Remained Undetected for over a Decade, Is Widespread in the USA"

_viruses, 2023, doi:10.3390/v15071597_

Round 1

Reviewer 1 Report

This study introduces a new virus, AmSV1, to the already diverse suite of RNA viruses that affect honey bees. The paper carefully and methodically proceeds to present the details of the virus extraction, sequence verification, and bioinformatics that were used to accurately identify and describe this new pathogen. The authors also describe the metagenomic virome analysis utilized which revealed a characteristic structure (the extended ORF that resembles other viruses from the same family). The experiments that were conducted are clearly presented and the historical data analysis is highly relevant.

The paper is extremely clean and well put together, I simply have a few questions for the authors, but excellent work! Fun to read! 

The phylogenetic study places this virus in the same clade as 2, already described, Solinivirus that infect ants but not in the same clade as a related Solinivirus that infects Apis cerana.

In the discussion you mention that the virus impacts egg laying in queen ants, which I think it’s fascinating, especially since the prevalence of the virus is highest during the growth season of the colony and obviously could impact new queens during swarming season. 

There is a mention, almost in passing of a related Solinivirus in A. cerana. Is there any evidence of detrimental impacts in A. cerana from this virus?

The authors proceed to describe the accumulation of AmSV1 in diverse areas of the body of individual worker honeybees (n=16) and present a solid case for high prevalence in the bee population.

There was a very significant correlation between head and thorax viral loads, but not so between head and abdomen, or between thorax and abdomen, But I am wondering, in Figure 2, there are four samples (1,2,3,6) that have a low viral load in the abdomen and those samples all have ND levels of AmSV1 on the head. In addition, sample 16 has ND levels for any of the 3 body parts. 

Those four samples constitute 25% (not including the negative sample) and abdomen viral loads from those samples seem to at or be below 103 GE which could hint to a threshold level before virus invades critical areas like the head. It’s possible that sixteen samples with great variability may obscure a possible relationship or threshold.

The impact of a viral infection, and the ability of the virus to replicate in a variety of tissues, could be influenced by the way is transmitted and the viral strain. Were any of the 16 samples analyzed for strain of AmSV1?

I have a few questions on the artificial inoculation of bee pupae with AmSV1.  1-why was 103 GE selected as the load to the used?, 2- was the virus extracted from whole bees? Why not use thorax or heads to exclude simple “digestive tract” viral positives? 3-was the parent colony from which these bee pupae were collected tested for presence of AmSV1 prior to pupal injections? I am curious because out of the 16 bees analyzed (head, thorax, abdomen) 15 were positive.

The historical analysis is a bit tricky because there were more samples from recent years compared to 2010 and many states did not get sampled until 2021, but I wish the authors would expand, if data permits, what may be happening in certain states such as TX (which changes from 2010 negative to 2014 positive), differences between places like ND and SD which were negative until 2014 and in 2021 ND still negative but SD is positive.  Is it simply  a sampling artifact?

It would also be interesting to consider what could happen to HI, a state where queen breeding is so important, and based on your data still tested negative in 2021. However, some HI queen breeders do import semen and if AmSV1 can be transmitted vertically the geographical isolation of Hawaii may be irrelevant. Sadly, if AmSV1 affects queen quality more efforts should be in place when semen importation to HI is involved.

Finally, I would probably only parasites for Varroa and Nosema only - not viruses which are included in that table.

Table S2. Connections between the prevalence of AmSV1 and other honey bee parasites

Author Response

Author's Reply to the Review Report (Reviewer 1)

Comments and Suggestions for Authors

This study introduces a new virus, AmSV1, to the already diverse suite of RNA viruses that affect honey bees. The paper carefully and methodically proceeds to present the details of the virus extraction, sequence verification, and bioinformatics that were used to accurately identify and describe this new pathogen. The authors also describe the metagenomic virome analysis utilized which revealed a characteristic structure (the extended ORF that resembles other viruses from the same family). The experiments that were conducted are clearly presented and the historical data analysis is highly relevant.

ANSWER: We thank Reviewer #1 for the constructive comments. We thought that it would be important, in addition to reporting novel virus sequence,  to carry out analysis of the incidence of this newly discovered virus,  and we are pleased that  Reviewer #1 appreciated our efforts.

The paper is extremely clean and well put together, I simply have a few questions for the authors, but excellent work! Fun to read! 

The phylogenetic study places this virus in the same clade as 2, already described, Solinivirus that infect ants but not in the same clade as a related Solinivirus that infects Apis cerana.

In the discussion you mention that the virus impacts egg laying in queen ants, which I think it’s fascinating, especially since the prevalence of the virus is highest during the growth season of the colony and obviously could impact new queens during swarming season. 

ANSWER: Yes, we are planning to investigate impact of AmSV1 infection on honey bee egg laying considering what is the negative effect of another Solinvivirus, SINV3 on its ant host ovaries and egg laying. We discuss this in L. 455-462.

There is a mention, almost in passing of a related Solinivirus in A. cerana. Is there any evidence of detrimental impacts in A. cerana from this virus?

ANSWER: There were no studies of incidence, or biological properties of Apis picorna-like virus 4 (GenBank Accession MZ822083), a solinvivirus found by metagenomic analysis of  Apis cerana. Therefore we could only mention that another Apis Soinvivirus was reported in China in A, cerana (L.254-256). 

The authors proceed to describe the accumulation of AmSV1 in diverse areas of the body of individual worker honeybees (n=16) and present a solid case for high prevalence in the bee population.

There was a very significant correlation between head and thorax viral loads, but not so between head and abdomen, or between thorax and abdomen, But I am wondering, in Figure 2, there are four samples (1,2,3,6) that have a low viral load in the abdomen and those samples all have ND levels of AmSV1 on the head. In addition, sample 16 has ND levels for any of the 3 body parts. 
Those four samples constitute 25% (not including the negative sample) and abdomen viral loads from those samples seem to at or be below 103 GE which could hint to a threshold level before virus invades critical areas like the head. It’s possible that sixteen samples with great variability may obscure a possible relationship or threshold.

The impact of a viral infection, and the ability of the virus to replicate in a variety of tissues, could be influenced by the way is transmitted and the viral strain. Were any of the 16 samples analyzed for strain of AmSV1?

I have a few questions on the artificial inoculation of bee pupae with AmSV1.  

1-why was 103 GE selected as the load to the used?, 

ANSWER: We injected 10^3 GE of virus because the yield was not very high. We had very limited number of adult worker in a 2021 frozen sample which was AmSV1-positive, therefore we  could use only 10 bees. Also,  two rounds of ultracentrifugation (including one through 20% sucrose), and filtration through 0.21 microns filter, inevitably reduced particle yield due to losses at every step. We obtained 500uL of partially purified virus prep (preparation is outlined in L.110-121) with a concentration of 3*10^3 AmSV1 genome equivalents per microliter. We used 1 microliter for injection (L 122-126).

2- was the virus extracted from whole bees? 

ANSWER: Yes, Yes, we used whole bees to obtain virus preparation used in the injection study (L.123)  "Individual bees (n=10)" changed to "Entire individual bees (n=10) "

Why not use thorax or heads to exclude simple “digestive tract” viral positives? 

ANSWER: Thank you for your suggestion. We had limited number of frozen bees, about 40 worker individuals, which had to be used for virus preparation and for analysis of virus loads in different body parts. Therefore we had to be very careful use limited amount of AmSV1 infected bee material. We specifically tested virus levels in different body parts. (Fig 2)

3-was the parent colony from which these bee pupae were collected tested for presence of AmSV1 prior to pupal injections? I am curious because out of the 16 bees analyzed (head, thorax, abdomen) 15 were positive.

ANSWER: Bees from the colony used for injection  were AmSV1-free. We mentioned this in (L. 133-136) "Honeybee pupae at the early pink eye stage obtained from a Florida colony which was free of AmSV1.." Also,  we included buffer-injection control which showed that the bees which did not receive AmSV1 were free of this virus ((Fig. 2). 

The historical analysis is a bit tricky because there were more samples from recent years compared to 2010 and many states did not get sampled until 2021, but I wish the authors would expand, if data permits, what may be happening in certain states such as TX (which changes from 2010 negative to 2014 positive), differences between places like ND and SD which were negative until 2014 and in 2021 ND still negative but SD is positive.  Is it simply  a sampling artifact?

ANSWER: Thank you for the comment. In our study we aimed to look at the current distribution of AmSV1 (which was done by testing all 930 apiary-levels bee RNA samples  collected as part of the USDA-APHIS National Honey Bee Disease Survey in 2021 (L. 133-136). We also tested a number of bee samples from 2010 and 2014 to find out if AmSV1 was already present or it was a recent arrival. The lack of detection in TX in 2010 could be due low number of sampled apiaries. We are planning future work to determine distribution of AmSV1 in different areas, including TX. 

It would also be interesting to consider what could happen to HI, a state where queen breeding is so important, and based on your data still tested negative in 2021. However, some HI queen breeders do import semen and if AmSV1 can be transmitted vertically the geographical isolation of Hawaii may be irrelevant. Sadly, if AmSV1 affects queen quality more efforts should be in place when semen importation to HI is involved.

ANSWER: Thank you for suggestion, we are planning future work to determine distribution of AmSV1 in different states, including HI. It is also in our plans to investigate routes of transmission of AmSV1. This paper focused on presenting sequence of AmSV1 , testing its replication in honey bees, and assessing its current distribution ( with possible connections with other parasites - microbes , varroa, and hive conditions). 

Finally, I would probably only parasites for Varroa and Nosema only - not viruses which are included in that table.
Table S2. Connections between the prevalence of AmSV1 and other honey bee parasites

ANSWER: Viruses also could be considered "parasites" (e.g from Miller & Krijnse-Locker (2008) Modification of intracellular membrane structures for virus replication, 6: 363–374, Abstract: Viruses are intracellular parasites that use the host cell they infect to produce new infectious progeny...). We used the most general term.

Reviewer 2 Report

I enjoyed this manuscript, the discovery of this novel virus is convincing and has been tested logically with nice additions of injection experiments and tissue detections. Combined with the prevalence data and associations with other pathogens and health metrics, the authors provide an interesting first study of this virus which may be important in honey bee health. Before recommending this manuscript be accepted for publication I would like to see some minor issues addressed. Largely, these are in reporting of the methods. I believe the methods section should comprehensively describe what was done in sufficient detail to be repeatable. Currently the methods fall far short of this. In particular:

-          How many bees/pool (approx.?)

-          How was QC performed – what software? Were samples excluded?

-          L78 – repetiion of ‘were’

-          Did you use BLAST manually for each sample? Or Blast2go etc? Were there cutoffs for not including contigs of a certain length? What reference database did you use? Nr? Virus db?

-          “which allowed the identification of sequences of the novel virus which …” – more info is needed here, how did you decide this was a novel virus?

-          For the protein seq alignment section – which protein? How did you decide what other sequences to include? What kind of phylogeny? Was there model seclection?

-          L111-114 – this sentence has unusual grammar

-          The RTqPCR details are lacking – which primers? SYBR green? Conditions? Housekeeping gene etc?

-          “Also, the virus levels were quantified in honey bee pupae injected with viral inoculum” This seems out of place in with the survey data

-           

In the results you show analysis of Shannon’s diversity and polymorphism calculations which didn’t appear in the methods, or negative strand detection. How many pupae did you use for the injections? There are different numbers of points on the plot in Fig3a to lanes on the gel in Fig3b.

 L 234 – “It was found that AmSV1 was grouped in the same clade as both classified solinviviruses, SINV3 and NfV1, with 96% bootstrap support.”  - While this is true, it’s not clear to me how robust this is. 9/12 viruses in your tree are in this same clade and it’s not clear what virus families the other viruses are in, or how you made the tree/chose the other sequences

 Finally

Were there other viruses (in original RNAseq data)? I know you found a particularly interesting contig/novel virus, but I wonder did you find other viruses in the original colonies with history of losses? Was there just the one potential new virus contig?

excellent. reads well, just a couple of instances of awkward grammar.

Author Response

I enjoyed this manuscript, the discovery of this novel virus is convincing and has been tested logically with nice additions of injection experiments and tissue detections. Combined with the prevalence data and associations with other pathogens and health metrics, the authors provide an interesting first study of this virus which may be important in honey bee health. 

ANSWER: We thank Reviewer #2 for the constructive comments. We thought that it would be important, in addition to reporting novel virus sequence,  to investigate prevalance of the newly discovered virus,  and we are very pleased that Reviewer #2 appreciated our effort.

Before recommending this manuscript be accepted for publication I would like to see some minor issues addressed. Largely, these are in reporting of the methods. I believe the methods section should comprehensively describe what was done in sufficient detail to be repeatable. Currently the methods fall far short of this. In particular:

-          How many bees/pool (approx.?)

ANSWER: L. 144. Added " from pools of 50 worker bees"

-          How was QC performed – what software? Were samples excluded?

ANSWER: L73. QC and adapter streaming was carried out by Genomics Resource Center (GRC), University of Maryland, Baltimore, MD. Only trimmed reads which passed QC (a standard procedure done by the Genomics Resource Center (GRC))  were released to us for further analysis (bowtie alignment, de novo assembly).

-          L78 – repetiion of ‘were’
ANSWER: Replaced for "and" (" ... Open Reading Frames (ORF) were analyzed and compared.." )

-          Did you use BLAST manually for each sample? Or Blast2go etc? Were there cutoffs for not including contigs of a certain length? What reference database did you use? Nr? Virus db?

ANSWER: There were only few long de novo assembled contigs, therefore BLAST of the contigs and the proteins encoded by contigs' ORFs was done manually. Additional information is provided, L. 80 - 85) 

-          “which allowed the identification of sequences of the novel virus which …” – more info is needed here, how did you decide this was a novel virus?

ANSWER: We extended Materials and method section to address this question:  (L. 70-83) "The contigs longer than 6 to 10.6 kb and their Open Reading Frames (ORF) were analyzed and compared against sequences deposited to GenBank by the BLAST search tool [14]. The search for similarities was done using tblastn, within NCBI nr/nt database, the cutoff E-value of 0.0 was used. This search showed that long ORF of some of some con-tigs coded for novel proteins showing 17% to 34% similarity to the proteins of the rec-ognized and putative members of the family Solviniviridae, order Picornavirales. " 

-          For the protein seq alignment section – which protein? How did you decide what other sequences to include? What kind of phylogeny? Was there model seclection?

ANSWER: We included proteins only proteins of the recognized and putative members of the family Solviniviridae, order Picornavirales. WE extended Materials and method section to address this question (L. 83-86) " The search for similarities was done using tblastn, within NCBI nr/nt database, the cutoff E-value of 0.0 was used. This search showed that long ORF of some of some contigs coded for novel proteins showing 17% to 34% similarity to the proteins of the recognized and putative members of the family Solviniviridae, order Picornavirales. 

-          L111-114 – this sentence has unusual grammar

ANSWER: rearranged this sentence to make it easier to read" (L 119-121) "Virus purification and pupal injection experiment. To obtain partially purified preparations of AmSV1, we used adult worker honey bees from the apiary samples, which had tested positive for AmSV1, were collected in an apiary in New York state in the summer 2021 and were stored at -80oC. "

         The RTqPCR details are lacking – which primers? SYBR green? Conditions? Housekeeping gene etc?

ANSWER: Primers are included in the Supplementary Table S1. See L 87-91 ("A set of oligonucleotide primers designed according to the AmSV1 contig (Supplementary Table S1) was used to produce a series of overlapping RT-PCR fragments covering the viral genome, using the RNA extract in which AmSV1 was discovered. The RT-PCR fragments were sequenced using the Sanger method and the confirmed sequence was submitted to GenBank under accession number OQ540582 (Supplementary Text S1). ")

-          “Also, the virus levels were quantified in honey bee pupae injected with viral inoculum” This seems out of place in with the survey data. 

ANSWER: The same AmSV1 qPCR primers were used for the US-wide survey (Fig, 4) and for quantification of AmSV1  in inoculation (Fi. 3a) and  the body section quantification experiment (Fig. 2b). So it is justified to mention this. 

-           In the results you show analysis of Shannon’s diversity and polymorphism calculations which didn’t appear in the methods,

ANSWER: Reference to Shannon'ss diversity calculation added L 96-100. "The AmSV1 sequence produced in our study, GenBank accession number OQ540582, was used as a reference sequence with Bowtie2 alignment tool [12] and samtools [17], the samtools mpileup output was used to calculate Shannon’s diversity essentially as in Ryabov et al. 2017 [18].  "

 or negative strand detection. 

ANSWER: The primers were described in L. 170-176 " For detection of the negative-strand RNA of AmSV1, total RNA extracted from honey bees was to produce cDNA with the tagged forward primer corresponding to the re-gion 5541 – 5562 in positive polarity (5’-CTTGGTTAGCTGTGTTGCAGTTGCATTTACAACGAACCAGTAAC-3’, the tag se-quence is underscored), then the resulted cDNA was used as a template in PCR reac-tion with the primer targeting the tag (5’-CTTGGTTAGCTGTGTTGCAGTTG-3’) and the reverse primer AmSV1-Rev. "

How many pupae did you use for the injections? There are different numbers of points on the plot in Fig3a to lanes on the gel in Fig3b.

ANSWER: Injection involved 8 pupae injected with AmSV1 virus preparation and 2 with PBS. For negative strand detection, pools of two pupae were made for the AmSV-injected. All is explained in Figure 3 and the legends.  

L 234 – “It was found that AmSV1 was grouped in the same clade as both classified solinviviruses, SINV3 and NfV1, with 96% bootstrap support.”  

- While this is true, it’s not clear to me how robust this is. 9/12 viruses in your tree are in this same clade and it’s not clear what virus families the other viruses are in, or how you made the tree/chose the other sequences

ANSWER. The alignment included only full length protein sequences of recognized and putative solinviviruses, we did not investigate phylogenetic relationships with other Picorna-like viruses because we used full length polyproteins solinviviruses and putative solinviviruses.  We explained that AmSV1 showed the highest similarity with solinviviruses in the text (.L 196-200)

 Finally

Were there other viruses (in original RNAseq data)? I know you found a particularly interesting contig/novel virus, but I wonder did you find other viruses in the original colonies with history of losses? Was there just the one potential new virus contig?

ANSWER: For assembly we removed reads corresponding to all known honey bee viruses and honey bee transcriptome. (Described in Therefore ably contigs the were assembled were of AmSV1. (L 79-83). 
"The reads were checked for quality and trimmed by Genomics Resource Center prior being mapped to the Apis mellifera honey bee transcriptome (OGSv3.2) [11] and the sequences of all known viruses infecting honey bees and Varroa mites with bowtie2 [12]. We detected reads belonging to Deformed wing virus-A, Deformed wing virus-A, Sac-brood virus, Black_queen cell virus, Israeli acute paralysis virus Apis_rhabdovirus-1, Apis rhabdovirus-2, Lake Sinai virus.  The un-aligned reads were then used for de novo assembly using Spades software [13]. "

There were other viruses

 ANSWER: we detected number the viruses: L. 76-79 "We detected reads belonging to Deformed wing virus-A, Deformed wing virus-A, Sac-brood virus, Black_queen cell virus, Israeli acute paralysis virus Apis_rhabdovirus-1, Apis rhabdovirus-2, Lake Sinai virus." Full details of this will be published in the paper Ref 10, which is now in preparation (Ref 10. Steinhauer et al). 

Reviewer 3 Report

The manuscript is well written and provide significant information related to honey bee viral infections. Moreover, as the authors express, the diagnostic method used would be usesfull to investigate the posible the connection of AmSV1 with honey bee colony losses. 

Author Response

The manuscript is well written and provide significant information related to honey bee viral infections. Moreover, as the authors express, the diagnostic method used would be useful to investigate the possible the connection of AmSV1 with honey bee colony losses. 

ANSWER: We are very pleased that Reviewer #1 likes our paper. Thank you for careful reading of our manuscript.